# Characterizing Edible Oils by Oblique-Incidence Reflectivity Difference Combined with Machine Learning Algorithms

**DOI:** 10.3390/foods13091420

**Published:** 2024-05-06

**Authors:** Xiaorong Sun, Yiran Hu, Cuiling Liu, Shanzhe Zhang, Sining Yan, Xuecong Liu, Kun Zhao

**Affiliations:** 1College of Computer and Artificial Intelligence, Beijing Technology and Business University, Beijing 100048, China; sunxr@th.btbu.edu.cn (X.S.); hyr18801449369@163.com (Y.H.); zhangsz@btbu.edu.cn (S.Z.);; 2Beijing Key Laboratory of Big Data Technology for Food Safety, Beijing Technology and Business University, Beijing 100048, China; 3College of Information Science and Engineering/College of Artificial Intelligence, China University of Petroleum, Beijing 102249, China; liuxc@student.cup.edu.cn; 4College of New Energy and Materials, China University of Petroleum, Beijing 102249, China; 5Key Laboratory of Oil and Gas Terahertz Spectroscopy and Photoelectric Detection, Petroleum and Chemical Industry Federation, China University of Petroleum, Beijing 102249, China

**Keywords:** oblique-incidence reflectivity difference, edible oils, machine learning, feature importance scores

## Abstract

Due to the significant price differences among different types of edible oils, expensive oils like olive oil are often blended with cheaper edible oils. This practice of adulteration in edible oils, aimed at increasing profits for producers, poses a major concern for consumers. Furthermore, adulteration in edible oils can lead to various health issues impacting consumer well-being. In order to meet the requirements of fast, non-destructive, universal, accurate, and reliable quality testing for edible oil, the oblique-incidence reflectivity difference (OIRD) method combined with machine learning algorithms was introduced to detect a variety of edible oils. The prediction accuracy of Gradient Boosting, K-Nearest Neighbor, and Random Forest models all exceeded 95%. Moreover, the contribution rates of the OIRD signal, DC signal, and fundamental frequency signal to the classification results were 45.7%, 34.1%, and 20.2%, respectively. In a quality evaluation experiment on olive oil, the feature importance scores of three signals reached 63.4%, 18.9%, and 17.6%. The results suggested that the feature importance score of the OIRD signal was significantly higher than that of the DC and fundamental frequency signals. The experimental results indicate that the OIRD method can serve as a powerful tool for detecting edible oils.

## 1. Introduction

Edible oils play an important role in our daily life, such as providing essential fatty acids, vitamins, and health-promoting ingredients [1,2]. The annual consumption of edible oil is large, which speaks to the importance of edible oil safety. Due to the influence of raw materials, there are significant price differences among different types of edible oils. Moreover, differences in brands, sources of raw materials, and processing techniques result in variations in ingredients and prices. Unscrupulous profit motives have led some businesses to deceive consumers by selling substandard products. Therefore, the safety testing of edible oil is of great importance in food safety assessment [3,4]. Common practices currently include blending soybean oil into olive oil, blending rapeseed oil into peanut oil, and so on [5,6]. When inferior-quality oil is mixed into edible oil, there may be issues such as exceeding aflatoxin limits and containing rancid fats. Consuming these oils can harm the nervous and digestive systems in the human body, and in severe cases, may even lead to cancer [7,8].

There are several procedures for researching edible oil in the field of food safety, such as flash gas chromatography [9,10,11], TaqMan real-time quantitative polymerase chain reaction [12], gas chromatography ion mobility spectrometry [13,14], electrochemical impedance spectroscopy [15], non-destructive ultrasonic method [16], and the use of chemometric tools [17]. Combined with chemometrics, flash gas chromatography had been used to assess volatile profiles of berry seed oils for authenticity and deterioration. The partial least squares model was used to predict storage time, and the results demonstrated that the coefficient of determination (R^2^) was between 0.842 and 0.969. Moreover, gas chromatography ion mobility spectrometry was employed to analyze secondary oxidation. A predictive model for the peroxide value of rapeseed oil was established in order to research the relationships between peroxide values and the contents of secondary oxidative products. In addition to this, a non-destructive ultrasonic method was used to investigate temperature-dependent acoustic parameters of edible oils. As the oil temperature ranged from 24 °C to 34 °C, acoustic parameters were closely related to the velocity, attenuation, and frequency components [16]. Using the HPLC-DAD method, the total phenolic content (TPC) and simple phenolic profile of raw olives were analyzed. The results indicated that Turkish olive varieties showed significant differences in phenolic characteristics due to variations in variety and harvest time [17].

Recently, rapid, collimated, and non-contact, optic methods have been widely used in edible oil detection. Fourier Transform Infrared spectroscopy was used to monitor the thermal stability of pure sesame oil, and a linear correlation was obtained between the FTIR signals at different conditions and the proportion of pure sesame oil, with the root mean square of prediction (RMSEP) between 0.8802 and 2.3827 and R^2^ between 0.9841 and 0.8834, respectively [18]. Moreover, surface-enhanced Raman spectroscopy, combined with an Artificial Neural Network, was used for the determination of peroxide value and fatty acid composition, with an accuracy of 99% [19]. In addition, fluorescence spectroscopy [20,21,22,23], reflectance spectroscopy [24,25,26], and terahertz spectroscopy [27,28,29,30] have also been used in edible oil measurement.

In this work, the oblique-incidence reflectivity difference (OIRD) method was proposed for edible oil detection. As an optical method, OIRD characterizes the surface properties of samples by the difference in reflectance values of p and s light after passing through the sample. Due to its advantage of non-destructive, fast, and high throughput, it is widely used in the fields of oil and gas reservoir exploration [31,32], as well as for monitoring biological chips [33,34] and map electronic transfer flux [35,36]. A prediction model was performed based on four different algorithms, which included Extreme Gradient Boosting, Logistic Regression, K-Nearest Neighbor, and Random Forest. The predicted results showed a clear correspondence with the content of monounsaturated fatty acids. Moreover, the contribution of OIRD signals was significantly higher than that of direct current signals and fundamental frequency signals. The experimental results suggested that OIRD is a useful tool in the detection of edible oil.

## 2. Materials and Methods

In this work, five types of edible oils, with six brands for each type, were used for testing. The information of the edible oil samples used is shown in Table 1. The first and second types of olive oil came from China, while the remaining four types of olive oil came from Spain. Corn oil, peanut oil, soybean oil, and rapeseed oil were all made in China, and all of them were non-genetically modified edible oils. Among them, olive oil 4, soybean oil 5, and peanut oil 3 were produced using cold pressing technology, while the rest were produced using physical pressing.

A schematic diagram of the experimental setup for OIRD is shown in Figure 1. A He-Ne laser was adopted with a power of 3.8 mW and a polarization ratio of 500:1. The laser operated at an incident angle corresponding to the Brewster angle (58°), with a 632.8 nm beam, the direction of the arrow is the direction of laser propagation. Moreover, incident light intensity was adjusted using an attenuator. The polarization degree was enhanced using a polarizer to modulate the laser into p-polarized light. The polarization state of the laser was further adjusted by a photo-clastic modulator to introduce s-polarized light at a frequency of 50 kHz. Thus, the s-polarized light and p-polarized light alternately exited the system. A quarter-wavelength phase shifter introduced a fixed phase difference between the two polarized components of the incident light. The reflected light was focused on the sample using a plano-convex lens. The reflected beam passed through an optical beam splitter, transformed into parallel light by another plano-convex lens, and then reached a silicon photodetector through a polarizer to suppress unwanted polarization. The signal was transmitted via a BNC cable to a lock-in amplifier for further processing. According to the Fresnel principle, when laser light is incident on a sample surface at a fixed angle (Brewster angle), the composition, structure, and density of the sample surface will affect the interface dielectric constant, thereby influencing the laser reflectance.

According to the Fresnel principle, when laser light is incident on the sample surface at a fixed angle (Brewster angle), the composition, structure, and density of the sample surface will affect the interface dielectric constant, thereby influencing the laser reflectance. The OIRD technique introduces two alternately emitted, mutually perpendicular linearly polarized lights (p and s) to the sample surface. It detects changes in the properties of the sample interface layer, such as thickness and dielectric constant. The Oblique incidence reflectivity difference is defined as follows:(1)$$\Delta p-\Delta s=\frac{r_{p}-r_{p0}}{r_{p0}}-\frac{r_{s}-r_{s0}}{r_{s0}}=\frac{\delta r_{p}}{r_{p0}}-\frac{\delta r_{s}}{r_{s0}}$$

Using the transfer matrix method allows for a quantitative analysis of the interaction process between light and matter, subsequently enabling the calculation of the reflection coefficients r_p_ and r_s_ for p-polarized and s-polarized light on the sample. By incorporating Equation (1), a quantitative expression for the Optical Interference Reflectance Difference (OIRD) signal in relation to the physical properties of the sample can be derived.
(2)$$\Delta p-\Delta s=d\frac{\left(\varepsilon_{d}-\varepsilon_{0})(\varepsilon_{d}-\varepsilon_{s}\right)}{\varepsilon_{d}}\frac{(-i)4\pi \sqrt{\varepsilon_{0}}\varepsilon_{s}\cos\varphi_{inc}\sin^{2}\varphi_{inc}}{\lambda (\varepsilon_{s}-\varepsilon_{0})\left(\varepsilon_{s}\cos^{2}\varphi_{inc}-\varepsilon_{0}\sin^{2}\varphi_{inc}\right)}$$

In this context, λ represents the wavelength of the incident laser, $\phi_{inc}$ denotes the incidence angle of the probing laser, and d signifies the thickness of the interface layer, while $\varepsilon_{0}$, $\varepsilon_{d}$, and $\varepsilon_{s}$ respectively represent the dielectric constants of the overlying layer, interface layer, and substrate. According to the Optical Interference Reflectance Difference (OIRD) monitoring mechanism, based on the direct current signal and the fundamental frequency signal (modulation frequency at 50 kHz) output by the lock-in amplifier, the difference in relative changes in laser reflectance can be obtained, as follows:(3)Im(Δp−Δs)≈12J1(π)I(Ω)IDC

When the modulation frequency is fixed, the x-th order Bessel function $J_{x}(A)$ becomes a constant. Combined with Equation (3), when the incident light wavelength λ and the incidence angle $\phi_{inc}$ are constant and the dielectric properties of the overlying layer and substrate are known, the Optical Interference Reflectance Difference (OIRD) technique can quantitatively detect interface thickness and dielectric properties.

To ensure the edible oil samples remained relatively stable over time, a single-point dynamic monitoring mode of the OIRD testing system was employed, and the estimated duration of each experimental test was set between 120 and 150 s. The output data were formatted as a 2n × 2 text document. The direct current signal I_DC_ and the fundamental frequency signal I(Ω = 50 kHz) were acquired through the lock-in amplifier. Subsequently, the OIRD signal was derived, and these three signals were employed as features for modeling analysis.

The physical properties were investigated by introducing laser into liquid samples and exploring the differences in reflectance values at the interface. However, scattering inevitably occurred in this process. Multi-scattering correction (MSC), as a data processing method, was designed to eliminate the influence of different scattering levels in the sample, which effectively enhanced data correlation and corrected the baseline shift and offset phenomena in the data by using ideal OIRD data. In this experiment, it was assumed that the average value of the OIRD data served as the ideal OIRD data.

The resistance of multiple-scattering correction to signal noise was limited. Thus, it could not completely eliminate the scattering noise in the data. Moreover, the OIRD signal was susceptible to external noise signals. A Savitzky–Golay (S-G) smoothing algorithm was suitable for data preprocessing. The S-G smoothing algorithm performs low-pass filtering on information to remove high-frequency components, effectively retaining low-frequency information [37]. Therefore, the noise was significantly suppressed.

In this work, four machine learning algorithms, including eXtreme Gradient Boosting (XGBoost), Random Forest (RF), Logistic regression (LR), and K-Nearest Neighbors (KNN), were employed in the data processing section to assist us in classification and feature importance scoring. XGBoost is an algorithm based on the Gradient Boosting Decision Tree (GBDT). In each iteration, GBDT learned a CART tree, fitting the difference between the predicted values of the preceding (t − 1) trees and the true values of the training set [38,39,40]. The process of generating trees in XGBoost is shown in Figure 2a. The results of weak classifiers trained by XGBoost were accumulated to obtain the final conclusion. The Random Forest algorithm combines Breiman’s “Boot-strap aggregating” idea with Ho’s “random subspace” method. RF is a classifier composed of multiple decision trees, and its output category was determined by the majority class among the individual tree outputs [41,42]. The basic principle of Random Forest is illustrated in Figure 2b. Logistic regression is a linear model derived from the exponential distribution family. It assumes that given input X, output Y follows a Bernoulli distribution. By introducing the Sigmoid function as a non-linear factor, logistic regression was widely used in classification problems [43,44]. By substituting the derivative of the Sigmoid function into the loss function of logistic regression, the gradient G was obtained, composed of partial derivatives. The process of gradient descent is described in Figure 2c. The K-Nearest Neighbors classifier is an online classifier that, during classification, identifies the K samples in the training set that are closest to the test sample and determines the class of the test sample based on these neighbors [45]. Figure 2d shows the flowchart of the KNN algorithm.

## 3. Results and Discussions

The OIRD time-domain signal is described in Figure 3. The signals of each sample were relatively smooth and there were significant differences in the OIRD signals of different edible oil samples. For corn oil 1, the OIRD signal Im(Δp − Δs) ranged from 0.2878 to 0.288. The Im(Δp − Δs) of corn oil 2 changed from 0.2851 to 0.2853. Moreover, the Im(Δp − Δs) of olive oil 1, peanut oil 1, rapeseed oil 1, and soybean oil 1 fluctuated around 0.2882, 0.2869, 0.2843, and 0.2864, respectively. However, it was difficult for us to distinguish all the oil samples based on the absolute magnitude of the signal values.

The average imaginary signals Im(Δp − Δs) are described in Figure 4. Except for the significantly lower OIRD signals of olive oil samples, it was difficult to distinguish different edible oil samples. This may be attributed to the considerably higher content of monounsaturated fatty acids in olive oil compared to others, leading to a lower dielectric constant in the interface layer of olive oil [46], consequently exhibiting a lower OIRD response. The fatty acid contents of different edible oils are shown in Table 2.

The average OIRD signal was taken as the ideal OIRD signal, and a multivariate scatter correction was applied to all OIRD data. This correction involved baseline shift and offset correction of the data based on the ideal OIRD data. Subsequently, the data after multivariate scatter correction were subjected to the S-G smoothing process. After iteratively comparing different parameter combinations for the smoothing model, the polynomial order and the number of smoothing points were set as 7 and 299 in order to achieve the best smoothing effect. Preprocessing effectively eliminated the influence of different scattering levels in the samples and removed external high-frequency noise. Figure 5a–e respectively displays the OIRD signals of corn oil, olive oil, peanut oil, rapeseed oil, and soybean oil after preprocessing.

The DC signal, fundamental frequency signal, and OIRD signal were selected as features in the experiment. The experiment adopted single-point dynamic scanning, and the continuous signal collection for each sample lasted for 120–150 s. To ensure the reliability of the results, six oil samples from different origins and brands were collected for each type of edible oil. Finally, approximately 50,000 stable sample points were selected for each type of edible oil as the dataset, in which 35,000 sample points were used for training, and the others were used for prediction. The model parameters used for processing OIRD data were described as follows. For the XGBoost, RF, and LR models, the random state was set as 2022. For the XGBoost model, the max depth, estimators, and verbosity were established as 6, 100, and 0, respectively. However, the parameter neighbor was set as 3 for the KNN model. Confusion matrices and prediction results are described in Figure 6 and Figure 7. It can be observed that except for LR, the accuracy of the other models in predicting the types of edible oils exceeds 95%. The lower accuracy of the LR model may be attributed to the complex relationships and interactions present in the interface layer dielectric constant and interface layer thickness, which are not simply linear. Additionally, the LR model exhibited limitations in handling continuous and discrete features, which resulted in suboptimal classification performance.

The predictive performance of XGBoost among the four models was exceptionally good, with an accuracy exceeding 97% for all types of edible oils. As a gradient boosting algorithm, XGBoost improved model performance by ensemble learning from multiple decision trees. Then, non-linear relationships could be analyzed by the XGBoost algorithm in order to achieve excellent predictive results. Due to external noise and potential data loss in the test, the XGBoost algorithm was used to handle missing values, which enhanced the reliability of prediction outcomes. Feature importance analysis was conducted using the XGBoost algorithm, and the feature importance scores for the three signals are shown in Figure 8. The contribution rates of the OIRD signal, DC signal, and fundamental frequency signal to the classification results were 45.7%, 34.1%, and 20.2%, respectively. Both the DC signal and the fundamental frequency signal were optical intensity signals received by the photodetector and amplified by the lock-in amplifier, which contributed less to the model’s predictive accuracy. By contrast, the OIRD signal was calculated from the DC and fundamental frequency signals according to the principles of OIRD technology, which can be expressed by the following relationship equation:(4)$$\frac{1}{2J_{1}(\pi )}\left(\frac{I(\Omega )}{I_{DC}}\right)=\frac{(-i)4\pi d\sqrt{\varepsilon_{0}}\varepsilon_{s}\cos\varphi_{inc}\sin^{2}\varphi_{inc}}{\lambda (\varepsilon_{s}-\varepsilon_{0})\left(\varepsilon_{s}\cos^{2}\varphi_{inc}-\varepsilon_{0}\sin^{2}\varphi_{inc}\right)}\frac{\left(\varepsilon_{d}-\varepsilon_{0})(\varepsilon_{d}-\varepsilon_{s}\right)}{\varepsilon_{d}}$$

The results indicated that the sole use of either DC or fundamental frequency signals did not effectively characterize the samples. However, the OIRD signal could be derived from the ratio of the two signals. Under the conditions of a fixed incident angle and modulation frequency of a photoelastic modulator, the OIRD signal could reflect both the thickness and dielectric properties of the interface layer. For the three classification prediction models, the OIRD data demonstrated good guiding capability for predicting edible oil types, with an accuracy of over 95%. The absolute value of the OIRD signals did not directly affect the prediction results. The unprocessed DC and fundamental frequency signals had a low contribution to the model. This can characterize the complex interplay between these two aspects. Applying the interface properties to differentiate types of edible oils exhibited high accuracy, which could be a novel method for addressing this issue.

Using single-point dynamic scanning, DC, fundamental frequency, and OIRD signals were collected from six olive oil samples, which originated from different brands and regions. The XGBoost, LR, RF, and KNN models were employed for quality analysis of the six olive oil samples. Each sample underwent continuous signal acquisition for 120–150 s. For each type of olive oil, approximately 10,000 stable sample points were selected as the dataset, while 7000 sample points were used for training, and the others were used for prediction. Confusion matrices and prediction results are shown in Figure 9 and Figure 10. The best-predicted result was for Olive oil2, where all models achieved an accuracy exceeding 98%. By contrast, the lowest accuracy was observed for Olive oil3, which responded to all three models achieving less than 75% accuracy. Olive oil3 achieved the highest monounsaturated fatty acid content among the six brands of olive oil, corresponding to 79%. However, the lowest monounsaturated fatty acid content was 70%—this referring to Olive oil4. The monounsaturated fatty acid content was a crucial indicator for evaluating the quality of olive oil. Additionally, the content of monounsaturated fatty acids is an important indicator for differentiating between different cooking oils. For example, corn oil and sunflower oil contain 28%, 23%.

Therefore, it is suggested that the OIRD method, combined with machine learning algorithms, can characterize the quality of olive oil.

Except for Olive oil3, the prediction accuracy for olive oils exceeded 97%. The feature importance scores for the three signals are shown in Figure 11. The contribution rates of the OIRD, DC, and fundamental frequency signals to the classification results were 63.4%, 18.9%, and 17.6%, respectively. The feature importance score of the OIRD signal was significantly higher than that of the DC and fundamental frequency signals, indicating the feasibility of evaluating olive oil quality based on interface properties. OIRD signals played a crucial role in model training, which carried sample physical property information. OIRD signals show a good consistency with the prediction accuracy of the model and the content of monounsaturated fatty acid.

In this work, the OIRD method was firstly proposed to detect edible oil. Combined with machine learning algorithms, the OIRD method can realize the classification of edible oils, which is beneficial for the quality inspection of edible oils. The detection models were established by OIRD data. Thus, these models were suitable for classifying edible oils through OIRD detection. And the test results were beneficial for consumers to understand the types and origins of edible oils. The models were established from the OIRD signal, DC signal, and fundamental frequency signal. In principle, the model was also applicable for analyzing the detection results of other edible oils. However, it may be necessary to establish a new model in order to determine specific parameters of edible oils.

Recently, the OIRD technique has been widely used in monitoring the in situ growth of oxide films [47,48,49,50,51,52,53], the preparation of biochips [54,55,56,57,58,59,60,61,62,63], and the exploration of oil and gas resources [64,65,66,67,68,69]. Experimental results suggested that the OIRD method could characterize the spatially resolved electrochemical reversibility of a polyaniline thin film. The OIRD signal would rise as the electrochemical conversion from a completely reduced state to a partially oxidized state [48]. Moreover, the deterioration of the electrochemical reversibility led to a decrease in the OIRD signal. The OIRD method has also been used in the scanning of biomolecules, realizing the label-free detection of biological molecular interaction [61]. In addition to this, based on the characterization of wax precipitation, the detection curve of OIRD can reveal the wax formation process [64]. In this paper, OIRD signals were used to guide models for the identification of edible oils. XGBoost, LR, RF, and KNN algorithms were employed for quality analysis of the six olive oil samples. Olive oil2 got the best-predicted result, while the lowest accuracy was observed for Olive oil3. Moreover, OIRD, DC, and fundamental frequency signals exhibited different contribution rates to the classification results. Going forward, our work lays the groundwork for future efforts by researchers to use this work as the starting point for the application of OIRD in the characterization of edible oils. Compared to other atomic-level monitoring tools, OIRD has significant advantages in data acquisition and image scanning time. Therefore, OIRD had become a high-throughput screening tool for protein detection. Additionally, the OIRD method was also used in biochip building. Just last year, OIRD technology made significant breakthroughs; the detection speed of OIRD microscopes improved by an order of magnitude, making OIRD a high-throughput screening tool. Its software has significant advantages in data acquisition and image scanning time compared to other atomic-level monitoring tools [54]. By applying polystyrene (PS) evenly onto a standard glass slide, a porous monolayer of PS can enhance the sensitivity of Oblique-incidence reflectivity difference (OIRD) through optical interference enhancement and effective dielectric constant effects [70]. This demonstrates that OIRD, as an emerging detection system, has shown significant advantages, while there are still many aspects that can be further researched and improved.

In recent years, optic methods have been widely used in food detection. Due to the limitations of the experimental setup, experiments can only be conducted in the laboratory. It is necessary to consider improving equipment to achieve online monitoring. At present, only the detection of edible oil in a stationary state has been considered, and it is necessary to consider the impact of external regulation on edible oil. Additionally, the types of edible oils evaluated in research are limited, especially lacking assessments on a wider variety of high-end edible oils, such as safflower edible oil [71,72]; the detection methods for them are currently relatively scarce and have limitations.

## 4. Conclusions

In this study, the OIRD method was employed for the characterization of edible oils. Key features including the DC signal, fundamental frequency signal, and OIRD signal were used to construct prediction models, employing advanced algorithms such as XGBoost, LR, RF, and KNN. Remarkably, the prediction accuracies of the XGBoost, RF, and KNN models all surpassed 95%. In addition, the feature importance scores of the OIRD signal, DC signal, and fundamental frequency signal were 45.7%, 34.1%, and 20.2% and 63.4%, 18.9%, and 17.6%, respectively. Experimental results indicated that the OIRD signal played an important role for the establishment of edible oil detection models. These findings underscored the significance of the OIRD method as a valuable tool for the precise measurement of edible oils. This study highlights the potential of OIRD as a promising technique for enhancing the efficiency and accuracy of edible oil analysis, thereby advancing research and applications in the field of food science and technology.

## Figures and Tables

**Figure 1 foods-13-01420-f001:**
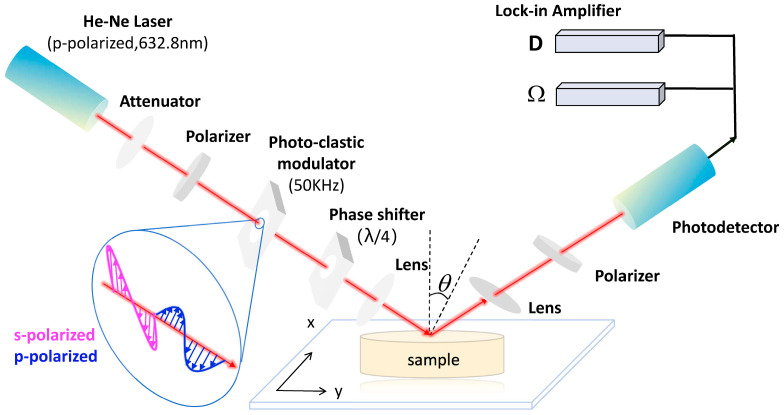
Schematic diagram of OIRD.

**Figure 2 foods-13-01420-f002:**
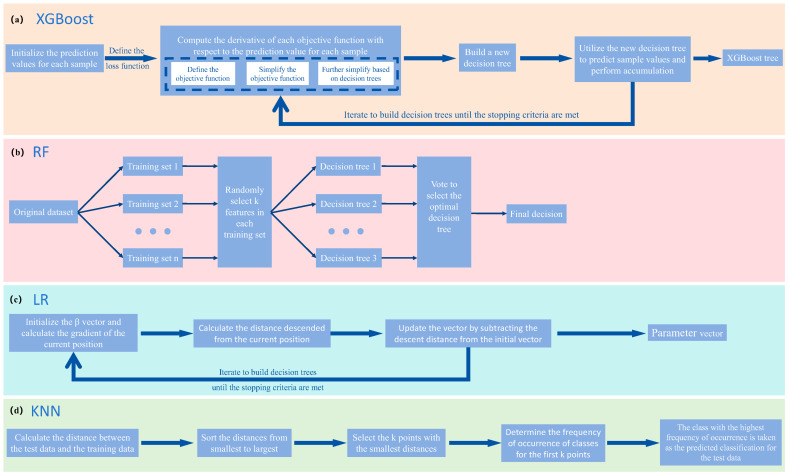
The schematic diagram of four machine learning algorithms.

**Figure 3 foods-13-01420-f003:**
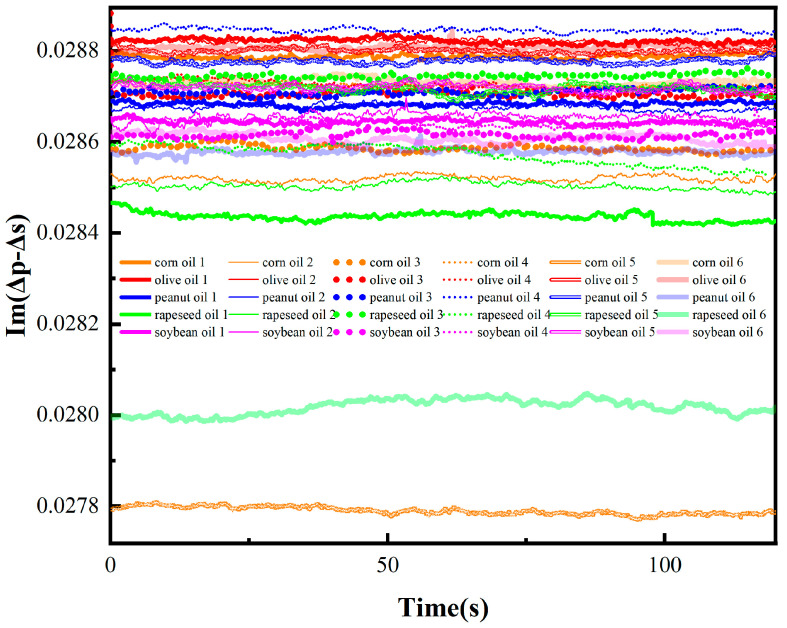
The temporal signals of OIRD for 30 edible oil samples.

**Figure 4 foods-13-01420-f004:**
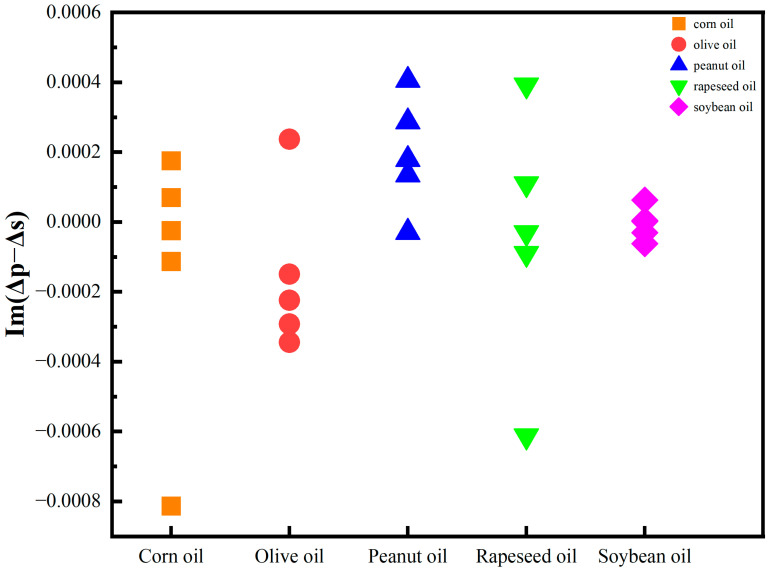
The average OIRD signal of edible oil samples.

**Figure 5 foods-13-01420-f005:**
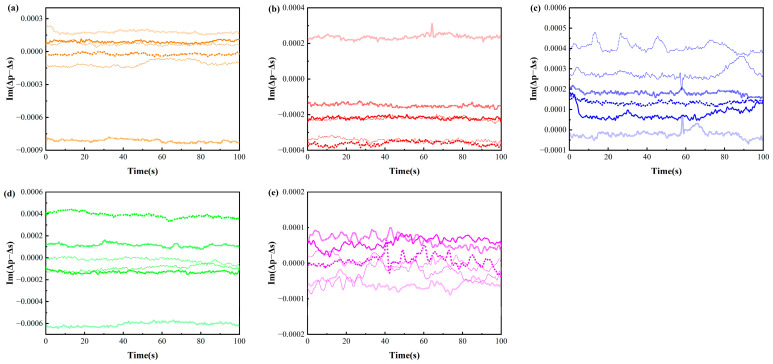
The OIRD temporal signals of five edible oils after preprocessing with S-G+MSC. (Different colours indicate different types of cooking oil, and different lines indicate different brands of cooking oil).

**Figure 6 foods-13-01420-f006:**
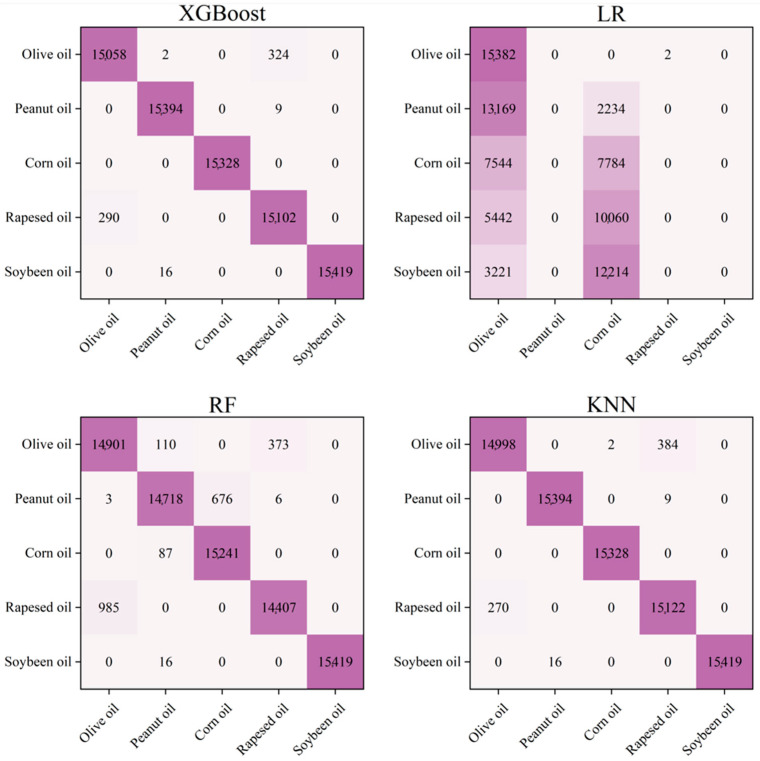
Confusion matrices for predicting the types of edible oils, using four models.

**Figure 7 foods-13-01420-f007:**
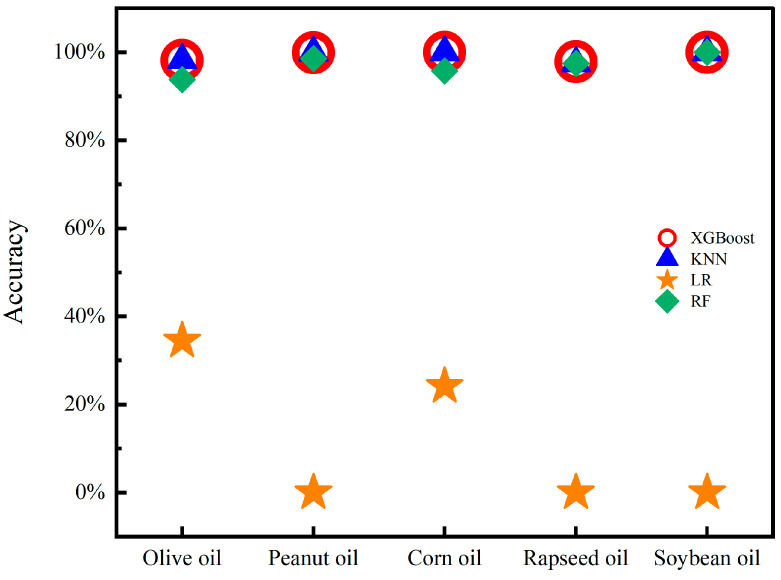
Accuracy of edible oil type prediction.

**Figure 8 foods-13-01420-f008:**
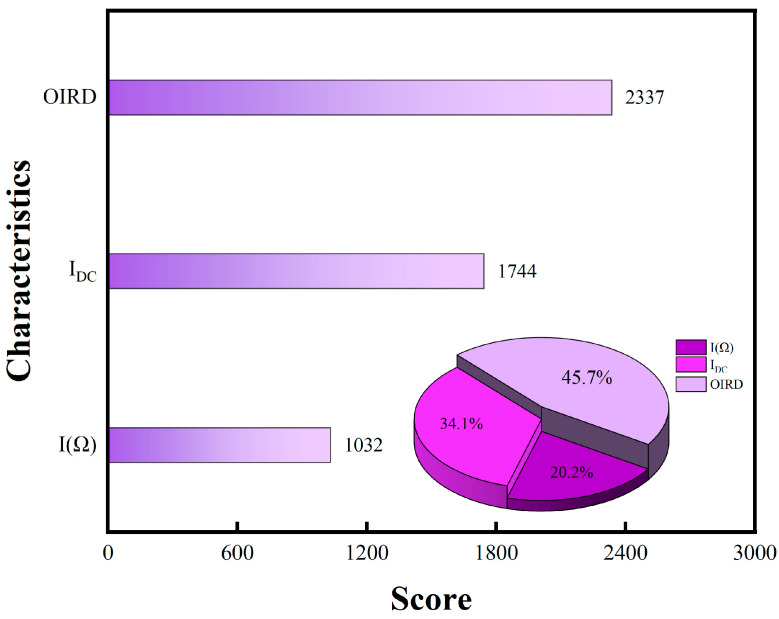
XGBoost feature importance scores for predicting the types of edible oils.

**Figure 9 foods-13-01420-f009:**
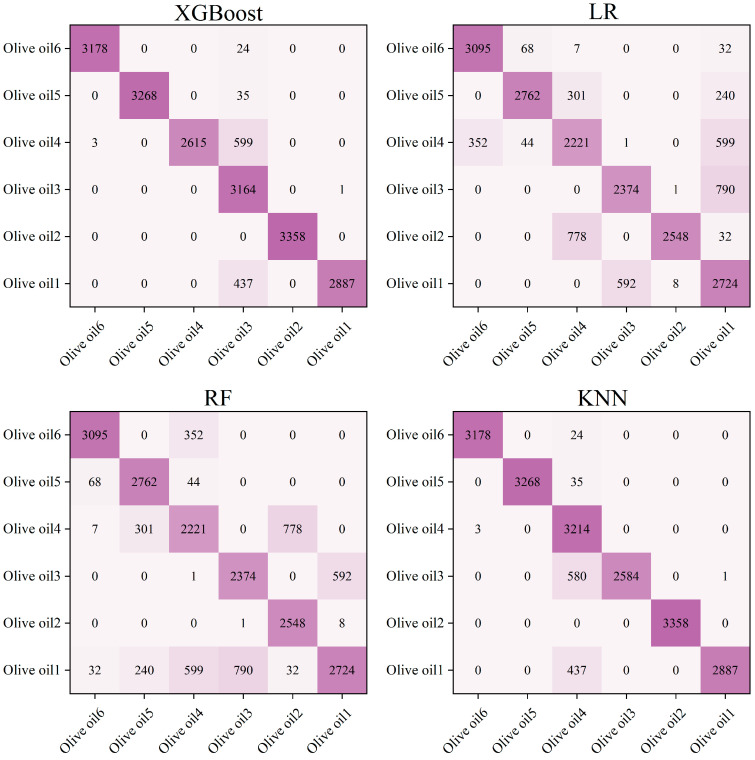
The confusion matrices for predicting the quality of olive oil using four models.

**Figure 10 foods-13-01420-f010:**
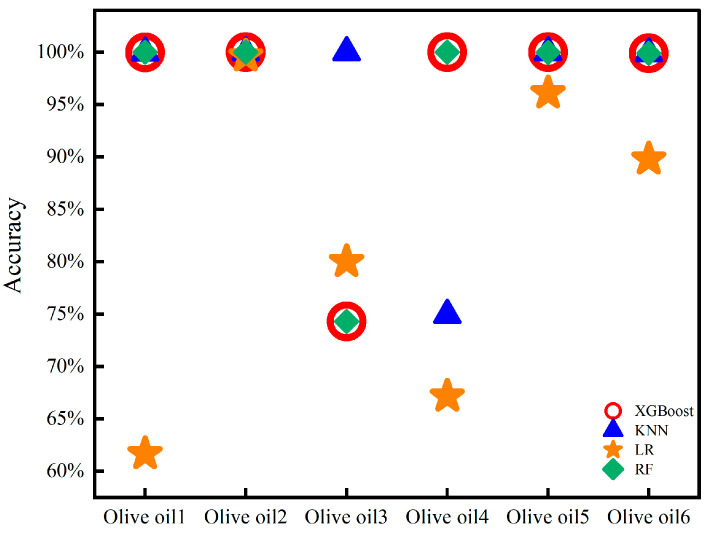
Accuracy of olive oil quality prediction.

**Figure 11 foods-13-01420-f011:**
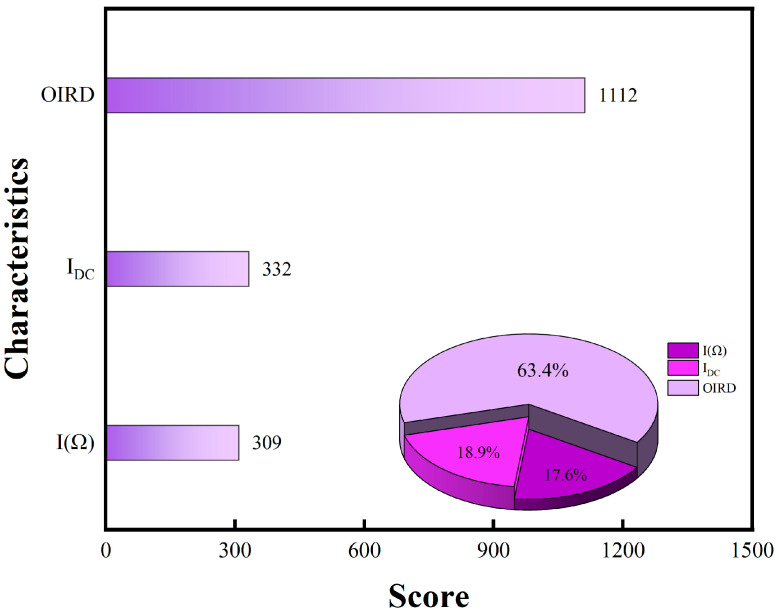
The feature importance scores for predicting the quality of olive oil using XGBoost.

**Table 1 foods-13-01420-t001:** Edible oil sample information.

Types of Edible Oils	Oil 1	Oil 2	Oil 3	Oil 4	Oil 5
corn oil	corn oil 1	corn oil 2	corn oil 3	corn oil 4	corn oil 5
olive oil	olive oil 1	olive oil 2	olive oil 3	olive oil 4	olive oil 5
peanut oil	peanut oil 1	peanut oil 2	peanut oil 3	peanut oil 4	peanut oil 5
rapeseed oil	rapeseed oil 1	rapeseed oil 2	rapeseed oil 3	rapeseed oil 4	rapeseed oil 5
soybean oil	soybean oil 1	soybean oil 2	soybean oil 3	soybean oil 4	soybean oil 5

**Table 2 foods-13-01420-t002:** The fatty acid content in edible oils.

Reference Table for the Fatty Acid Content in Edible Oils (%)				
Edible Oils	Saturated Fatty Acids	Monounsaturated Fatty Acids	Polyunsaturated Fatty Acids	
		Oleic Acid (Ω-9)	Linoleic Acid (Ω-6)	Alpha-Linolenic Acid (Ω-3)
corn oil	10–13	20–25	50–60	4–6
olive oil	9–11	67–75	10–15	0–3
peanut oil	17–18	30–40	30–40	0–3
rapseed oil	5–10	40–60	10–20	5–8
soybean oil	10–13	20–25	55–60	4–6

## Data Availability

The original contributions presented in the study are included in the article, further inquiries can be directed to the corresponding authors.

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
