# Peer review of "Characterizing Edible Oils by Oblique-Incidence Reflectivity Difference Combined with Machine Learning Algorithms"

_foods, 2024, doi:10.3390/foods13091420_

Round 1

Reviewer 1 Report

Comments and Suggestions for Authors

Dear editor of Food MDPI, first of all I want to thank you for the opportunity to improve the quality of manuscripts that are sent to your prestigious journal. I tell you that I have reviewed the manuscript titled "Characterizing edible oils by oblique-incidence reflectivity difference combined with machine learning algorithms " and I consider the reviewers make the following adjustments and comments that I include continuation. My decision is major revision.

Comment to authors

1. In the summary section include 2 examples of the most common cheap edible oils by which olive oil can be adulterated. Additionally, include 2 or 3 negative health impacts of consuming adulterated oils.

2. A Figure should be included in the materials and methods section that exemplifies how the application design of the four machine learning algorithms was carried out, including eXtreme Gradient Boosting (XGBoost), Random Forest (RF), Logistic regression (LR) and K- Nearest Neighbors (KNN), with its inputs and outputs.

3. In the results section, further discussion should be included in each determination carried out to substantiate that OIRD is a verified tool to determine adulteration of edible oils.

4. Make a section before the conclusion section that refers to the limitations of the study. Here I could include that more edible oils should be evaluated such as safflower edible oil due to its high commercial level worldwide and in countries such as India and northern Mexico due to its arid and semi-arid conditions.. Cite: (2021). Recovery of phytochemical from three safflower (Carthamus tinctorius L.) byproducts: Antioxidant properties, protective effect of human erythrocytes and profile by UPLCDADMS. Journal of Food Processing and Preservation45(9), e15765. (2021). Sustainable-green synthesis of silver nanoparticles using safflower (Carthamus tinctorius L.) waste extract and its antibacterial activity. Heliyon7(4).

5.  If I improved the research conclusion for its best impact, please include it:

In this study, the OIRD method was employed for the characterization of edible oils. Key features including the DC signal, fundamental frequency signal, and OIRD signal were used to construct prediction models, employing advanced algorithms such as XGBoost, LR, RF, and KNN. Remarkably, the prediction accuracies of the XGBoost, RF, and KNN models all surpassed 95%. Additionally, the OIRD signal demonstrated a notably higher contribution rate compared to the fundamental frequency signal and DC signal. These findings underscore the significance of the OIRD method as a valuable tool for the precise measurement of edible oils. This study highlights the potential of OIRD as a promising technique for enhancing the efficiency and accuracy of edible oil analysis, thereby advancing research and applications in the field of food science and technology.

 Dear Editor,

We would like to submit our latest manuscript entitled “Characterizing edible oils by oblique-incidence reflectivity difference combined with machine learning algorithms” by the following authors: Xiaorong Sun, Yiran Hu, Cuiling Liu, Shanzhe Zhang, Sining Yan, Xuecong Liu and Kun Zhao to Foods.

Due to the significant price difference, expensive oils like olive oil are often blended with cheaper edible oils, common practices currently include blending soybean oil into olive oil, blending rapeseed oil into peanut oil, and so on. This practice of adulteration in edible oils, aimed at increasing profits for producers, poses a major concern for consumers. Furthermore, adulteration in edible oils can lead to various health issues impacting consumer well-being, especially when inferior quality oil is mixed into edible oil, there may be issues such as exceeding aflatoxin limits and containing rancid fats. Consuming these oils can harm the nervous and digestive systems in the human body, and in severe cases, may even lead to cancer. In order to meet the requirements of fast, non-destructive, universal, accurate and reliable quality testing for edible oil, the oblique-incidence reflectivity difference (OIRD) method combined with machine learning algorithms was introduced to detect a variety of edible oils. The prediction accuracy of Gradient Boosting, K-Nearest Neighbor, and Random Forest models exceeded 95%.In the experiment of distinguishing different types of edible oils, the contribution rates of OIRD signal, DC signal, and fundamental frequency signal to the classification results were 45.7%, 34.1%, and 20.2%, respectively, In the quality evaluation experiment of olive oil, the feature importance scores of three signals reached 63.4%, 18.9%, and 17.6%, respectively. The feature importance scores of the OIRD signal was significantly higher than that of the DC and fundamental frequency signals, This indicates that the OIRD signal has a good guiding significance for the establishment of edible oil type and quality detection models.Experimental results indicate that the OIRD method can serve as a powerful tool for detecting edible oils.

The article is original and unpublished and is not being considered for publication elsewhere. If you have further request about this manuscript, please inform me by the communication information above.

The corresponding author was supported by the National Natural Science Foundation of China under Grant 12374412 and the Beijing Natural Science Foundation under Grant 4222043.

Thank you very much.

Reviewer 2 Report

Comments and Suggestions for Authors

1. The abstract should be revised and improved, more numirical data could be added, the main insights of this study should be added e.g., the feature importance scores for predicting the quality of olive oil using XGBoost, DC signal, fundamental frequency signal, and OIRD signal, etc.

2. Add the reference/s for L 27-34.

3. L 69: More details about these brands must be added, the name, origin, etc.. to reproduce this research.

4. The introduction is boor, and non connected, revise it, and add more relevant studies especially about chemometric tools and also about the detection of the adultration thru determining of biophenolic content of the main edible oils.

5. The discussion section must be improved with more studies and recent data.

Comments on the Quality of English Language

Dear Editor,

We would like to submit our latest manuscript entitled “Characterizing edible oils by oblique-incidence reflectivity difference combined with machine learning algorithms” by the following authors: Xiaorong Sun, Yiran Hu, Cuiling Liu, Shanzhe Zhang, Sining Yan, Xuecong Liu and Kun Zhao to Foods.

Due to the significant price difference, expensive oils like olive oil are often blended with cheaper edible oils, common practices currently include blending soybean oil into olive oil, blending rapeseed oil into peanut oil, and so on. This practice of adulteration in edible oils, aimed at increasing profits for producers, poses a major concern for consumers. Furthermore, adulteration in edible oils can lead to various health issues impacting consumer well-being, especially when inferior quality oil is mixed into edible oil, there may be issues such as exceeding aflatoxin limits and containing rancid fats. Consuming these oils can harm the nervous and digestive systems in the human body, and in severe cases, may even lead to cancer. In order to meet the requirements of fast, non-destructive, universal, accurate and reliable quality testing for edible oil, the oblique-incidence reflectivity difference (OIRD) method combined with machine learning algorithms was introduced to detect a variety of edible oils. The prediction accuracy of Gradient Boosting, K-Nearest Neighbor, and Random Forest models exceeded 95%.In the experiment of distinguishing different types of edible oils, the contribution rates of OIRD signal, DC signal, and fundamental frequency signal to the classification results were 45.7%, 34.1%, and 20.2%, respectively, In the quality evaluation experiment of olive oil, the feature importance scores of three signals reached 63.4%, 18.9%, and 17.6%, respectively. The feature importance scores of the OIRD signal was significantly higher than that of the DC and fundamental frequency signals, This indicates that the OIRD signal has a good guiding significance for the establishment of edible oil type and quality detection models.Experimental results indicate that the OIRD method can serve as a powerful tool for detecting edible oils.

The article is original and unpublished and is not being considered for publication elsewhere. If you have further request about this manuscript, please inform me by the communication information above.

The corresponding author was supported by the National Natural Science Foundation of China under Grant 12374412 and the Beijing Natural Science Foundation under Grant 4222043.

Thank you very much.

Reviewer 3 Report

Comments and Suggestions for Authors

In this paper, the characterization of edible oils was done using the oblique-incidence reflectivity difference (OIRD) method and a prediction model was obtained using machine learning. The article is well written, understandable and structured. However, some parts need to be supplemented in the paper. 

Concrete results are missing in the abstract and conclusion.

It is not common for the oblique-incidence reflectivity difference (OIRD) method to be used in the analysis of oils and fats. Add something more about the application of this method in the introduction part.

Model validation parameters that claim that the OIRD method is good for characterizing edible oils should be added.

No limitations of this study have been highlighted.

The practical significance of this research is not explained. Where could the resulting models be applied, who could they benefit from? Are these models obtained from 5 different oils sufficient for use with a larger range of different oils? Or is it necessary to make new models?

Author Response

Dear Editor,

We would like to submit our latest manuscript entitled “Characterizing edible oils by oblique-incidence reflectivity difference combined with machine learning algorithms” by the following authors: Xiaorong Sun, Yiran Hu, Cuiling Liu, Shanzhe Zhang, Sining Yan, Xuecong Liu and Kun Zhao to Foods.

Due to the significant price difference, expensive oils like olive oil are often blended with cheaper edible oils, common practices currently include blending soybean oil into olive oil, blending rapeseed oil into peanut oil, and so on. This practice of adulteration in edible oils, aimed at increasing profits for producers, poses a major concern for consumers. Furthermore, adulteration in edible oils can lead to various health issues impacting consumer well-being, especially when inferior quality oil is mixed into edible oil, there may be issues such as exceeding aflatoxin limits and containing rancid fats. Consuming these oils can harm the nervous and digestive systems in the human body, and in severe cases, may even lead to cancer. In order to meet the requirements of fast, non-destructive, universal, accurate and reliable quality testing for edible oil, the oblique-incidence reflectivity difference (OIRD) method combined with machine learning algorithms was introduced to detect a variety of edible oils. The prediction accuracy of Gradient Boosting, K-Nearest Neighbor, and Random Forest models exceeded 95%.In the experiment of distinguishing different types of edible oils, the contribution rates of OIRD signal, DC signal, and fundamental frequency signal to the classification results were 45.7%, 34.1%, and 20.2%, respectively, In the quality evaluation experiment of olive oil, the feature importance scores of three signals reached 63.4%, 18.9%, and 17.6%, respectively. The feature importance scores of the OIRD signal was significantly higher than that of the DC and fundamental frequency signals, This indicates that the OIRD signal has a good guiding significance for the establishment of edible oil type and quality detection models.Experimental results indicate that the OIRD method can serve as a powerful tool for detecting edible oils.

The article is original and unpublished and is not being considered for publication elsewhere. If you have further request about this manuscript, please inform me by the communication information above.

The corresponding author was supported by the National Natural Science Foundation of China under Grant 12374412 and the Beijing Natural Science Foundation under Grant 4222043.

Thank you very much.

Round 2

Reviewer 1 Report

Comments and Suggestions for Authors

The manuscript is accepted in its current form

Reviewer 2 Report

Comments and Suggestions for Authors

The manuscript has been revised and can be accepted after the language editing. 

Comments on the Quality of English Language

Should be improved.